# Psychosocial Representations of Gender-Based Violence Among University Students from Northwestern Italy

**DOI:** 10.3390/bs15101373

**Published:** 2025-10-08

**Authors:** Ilaria Coppola, Marta Tironi, Elisa Berlin, Laura Scudieri, Fabiola Bizzi, Chiara Rollero, Nadia Rania

**Affiliations:** 1Department of Educational Sciences, University of Genoa, 16128 Genoa, Italy; ilaria.coppola@edu.unige.it (I.C.); laura.scudieri@unige.it (L.S.); fabiola.bizzi@libero.it (F.B.); nadia.rania@unige.it (N.R.); 2Department of Psychology, University of Turin, 10124 Turin, Italy; elisa.berlin@unito.it (E.B.); chiara.rollero@unito.it (C.R.)

**Keywords:** young adults, university students, gender-based violence, awareness, qualitative research, perception, Italian study

## Abstract

The aim of the study was to explore the psychosocial perceptions that young adults have regarding gender-based violence, including those based on their personal experiences, and to highlight perceptions related to social media and how its use might be connected to gender-based violence. The participants were 40 university students from Northwestern Italy with an average age of 21.8 years (range: 19–25); 50% were women. Sampling was non-probabilistic and followed a purposive convenience strategy. Semi-structured interviews were conducted online and audio-recorded, and data were analyzed using the reflective thematic approach. The results revealed that young adults are very aware, at a theoretical level, of “offline” physical, psychological, and verbal gender-based violence and its effects, while they do not give much consideration to online violence, despite often being victims of it, as revealed by their accounts, for example, through unsolicited explicit images or persistent harassment on social media. Therefore, the results of this research highlight the need to develop primary prevention programs focused on increasing awareness and providing young people with more tools to identify when they have been victims of violence, both online and offline, and to process the emotional experiences associated with such events.

## 1. Introduction

Violence against women is defined as “any act of gender-based violence that results in, or is likely to result in, physical, sexual, or psychological harm or suffering to women, including threats of such acts, coercion, or arbitrary deprivation of liberty, whether occurring in public or private life” ([55]). It is recognized as a global social issue that not only exacerbates gender inequality but also has profound repercussions at both the individual and societal levels. On an individual level, numerous scholars have emphasized that violence against women has long-term and, in some cases, fatal consequences on women’s health and well-being ([44]; [51]). From a societal perspective, the progressive isolation of women, the lack of participation in daily life activities and employment, as well as a diminished ability to engage in self-care, result in significant social and economic costs that extend to society ([60]). Violence against women can take multiple forms: the most severe violation of women’s rights is sexual violence that refers to any act or attempt directed against a person’s sexuality, including rape ([11]), and can be perpetrated by anyone, regardless of their relationship with the victim ([59]). Another widespread form is intimate partner violence (IPV) that refers to a set of behaviors, expressed in various ways, perpetrated by a current or former intimate partner, leading to physical, sexual, and psychological harm. The literature has further highlighted that women who are victims of IPV have a higher risk of developing depression, anxiety, and post-traumatic stress disorder ([47]). According to data from the World Health Organization ([59]), regardless of cultural, religious, or economic background, women in any country can be subjected to violence. Globally, one in three women experiences physical and/or sexual violence, primarily by an intimate partner, while it is estimated that one in four women will experience physical and/or sexual violence by an intimate partner at some point in their lives. To date, 50 million women aged 18–74 in Europe experienced physical (including threats) or sexual violence ([23]).

On a national level, data from National Institute of Statistics indicate that 31.5% of women have experienced some form of physical or sexual violence during their lifetime. In 62.7% of rape cases, the perpetrator was the victim’s partner ([39]).

Recognizing the various forms of violence against women is of paramount importance to effectively address and combat this issue. Research over the years has shown that people tend to more easily recognize and consider physical violence as more severe compared to non-physical forms such as psychological and emotional violence ([45]). Both individual and social factors influence attitudes toward these forms of violence ([26]), with gender being one of the most significant. The literature indicates that men are more likely to endorse beliefs that justify violence against women, often underestimating the severity of certain violent behaviors directed at the female gender ([26]; [56]; [58]). Earlier perspectives interpreted gender roles and related violence as the result of biological differences, often linking male aggression to evolutionary predispositions or hormones such as testosterone ([31]). Since the 1970s, however, these deterministic views have been progressively replaced by socio-cultural approaches, which emphasize that gender is a socially constructed category shaped through norms and socialization processes ([15]; [48]). Through family, education, and peer interactions, norms of masculinity centered on dominance and control, and expectations of female subordination, are internalized across generations ([43]). These cultural constructions contribute to the normalization and justification of certain forms of violence, reinforcing unequal power relations.

### 1.1. Gender-Based Violence Among University Students: Perceptions and Cultural Configurations

The national and international literature highlights that individuals under the age of 29 are generally at a higher risk of experiencing gender-based violence ([22]; [27]). Therefore, understanding university students’ perception of this phenomenon is crucial, as the age at which individuals attend college coincides with the period when they are most at risk. Gender-based violence is a well-known issue within the university student community, although it is perceived with greater sensitivity and awareness by female students ([4]). Specifically, young people more easily identify physical violence, followed by sexual, psychological, and socio-economic violence, while they tend to perceive psychological violence as less severe ([37]; [50]). More specifically, they are more likely to recognize sexual violence when it occurs between strangers but tend not to acknowledge it within their own romantic or sexual relationships ([37]). Despite this, some studies indicate that the most prevalent form of violence in university settings is psychological abuse ([52]), along with the presence of sexual or sexist comments ([53]). These forms of violence have been found to be most strongly and negatively correlated with mental health and academic outcomes ([61]). Furthermore, research has shown that women are less likely than men to justify violence occurred in dating relationships, as revealed in studies investigating young people’s attitudes toward this issue ([24]).

Social media has become the primary mode of interaction, especially among younger generations, as shown by recent research indicating that university students spend several hours daily on these platforms, with significant implications for their well-being ([41]). Gender-based violence against women on social media represents a significant issue, as it is both widespread and harmful at the individual and societal level. It takes various forms, such as harassment, threats, and abuse, leading to psychological, emotional, and physical consequences for the victims ([17]). An analysis conducted by the Italian National Institute of Statistics shows that online violence originates from the same cultural and social roots as offline violence against women ([35]). A study with Spanish university students (18–22) reveals that social media often serve as instruments of control and harassment in relationships, normalizing gender-based violence. Psychological abuse is identified as the most prevalent form, while risks linked to sexting highlight the need for prevention strategies([30]).

Interactions and relationships among young adults today are profoundly influenced by the use of social media ([20]). However, studies indicate that the form of violence least recognized by this age group is cyber-violence (such as non-consensual intimate image, cyber stalking, cyber harassment), despite its increasing prevalence ([18]; [38]). In this regard, [18] ([18]) highlight the limited scope of research on this topic. Their study found that 48.1% of female participants had, at some point, received an unsolicited explicit image from an unknown man. The study conducted by [20] ([20]) highlights a reciprocal influence between jealousy expressed on social media—specifically regarding a partner’s social activities—and offline intimate partner violence, underscoring the risks associated with social media use in romantic relationships among young adults. The same authors emphasize that social media has profoundly transformed social interactions in this age group, particularly in the context of intimate relationships. While they can enhance relationship satisfaction—especially in long-distance relationships—they can also sometimes exacerbate pre-existing conflicts. Social media has thus enabled abusive behavior to transcend physical barriers, contributing to the shift in gender-based abuse from offline to online spaces ([32]). [32] ([32]) also notes that much of the literature has focused on how online violence has intensified forms of bullying, with particular attention to coercive sexting under pressure and the non-consensual distribution of sexually explicit images. [34] ([34]), however, stress the importance of differentiating between online and offline forms of violence in order to fully grasp their complexity and intersectionality. However, little is known about how these forms of violence manifest in dating and relationship contexts.

Several studies have shown that gender-based violence is linked to gender construction and how the categories of “men” and “women,” framed within a binary structure, are enacted about traditional gender roles and intimate practices ([3]; [28]; [62]). Although numerous studies have investigated university students’ beliefs and attitudes toward gender-based violence, both online and offline, through quantitative approaches ([4]; [18]; [42]; [53]) or have focused on gender-based violence within university settings ([5]; [57]), there remains a lack of research exploring how university students conceptualize and experience gender-based violence through qualitative approaches that center participants’ perspectives and lived experiences using an in-depth exploration. Despite the growing body of research on gender-based violence among university students, most studies have relied on quantitative methods or focused on institutional contexts. There remains a gap in understanding how young adults personally conceptualize and experience gender-based violence, particularly in relation to social media and gender identity. Understanding how university students conceptualize gender-based violence is crucial, as it allows us to identify which behaviors are recognized as violent and which, instead, tend to be normalized or minimized. It also highlights the role that gender roles and stereotypes play in shaping attitudes and relational practices among younger generations. Such knowledge provides the foundation for developing prevention and intervention programs within university curricula that are tailored to the specific needs expressed by students themselves.

To address this gap, we conducted a qualitative study aimed at exploring the psychosocial representations of gender-based violence among university students in Italy, with a focus on their lived experiences and perceptions.

### 1.2. The Research Aims

This study seeks to answer the following research questions:What psychosocial representations do university students hold regarding gender-based violence, including its definitions and perceived causes?Are there differences in representations and experiences based on gender? In what ways do representations and experiences differ between men and women?How is the use of social media perceived in relation to gender-based violence?

## 2. Materials and Methods

This study presents the data emerged from a semi-structured interview created ad hoc for this study that involved university students. The most evident advantage of this methodology is that it allows researchers to focus on specific topics while giving the interviewer a certain degree of autonomy in exploring relevant aspects that may emerge during the interview itself ([2]; [36]). [36] ([36]) define the semi-structured interview as an exploratory interview, a flexible measure that enables the investigation of specific topics whenever the participant is willing to provide more in-depth insights beyond those foreseen in the formalized research design.

The outline used for the interview included stimulus questions aimed at exploring representations of gender-based violence (e.g., What does gender-based violence mean to you? Tell me the first three words that come to your mind to describe gender-based violence), the psychosocial causes and consequences of gender-based violence (e.g., What do you think are the causes of these forms of violence?), and psychosocial representations of relationships with partners (e.g., Can you tell me three words to describe the relationship with your last (ex) partner? I refer to what it is like to be in this relationship) or experience about receiving intimate photo/video of another person (e.g., Have you ever privately received intimate photos/videos of another person? Did you expect them? What motivations do you think prompted the other person? How did you react to it? How did it make you feel?). Moreover, the interviewers were free to ask additional questions, consistent with the research objective, to further explore issues that emerged during each individual interview, allowing for a deeper understanding of participants’ personal perspectives. This made it possible to move from the theoretical representation of gender-based violence to the experiential level and, thus, answer the research questions: Are there differences in representations and experiences based on gender? In what ways do representations and experiences differ between men and women?

### 2.1. Procedures

The interviews were conducted by two members of the research team with a background in psychology. Both interviewers identify as female. All interviews were conducted online via the Teams platform and were audio-recorded. The platform automatically generated a verbatim transcript of the interview, which was then reviewed and cleaned by research assistants. Each interview lasted on average 45 min (range: 30–65). The final sample comprised 40 participants. Consistent with Braun and Clarke ([9]), our reflexive approach views themes as actively constructed through interpretative engagement with the data, rendering the concept of saturation as a fixed endpoint neither applicable nor coherent with our methodology. We selected a sample size that allowed for an in-depth exploration of participants’ experiences while ensuring sufficient diversity to develop robust and rich themes. Throughout the analysis, ongoing reflexivity and dialog among researchers ensured that the developed themes comprehensively addressed our research questions. Rather than aiming for a definitive saturation point, we prioritized interpretative adequacy and methodological coherence. The interviews were voluntarily attended by students from two University of Genoa and Turin, regardless of their degree program. The only eligibility criteria were that participants were between 19 and 25 years old and enrolled in a university program. Sampling was non-probabilistic and followed a purposive convenience strategy, recruiting volunteers through university mailing lists and social media announcements. This selection criterion was chosen in order to reach university students coming from different university environments, so as to bring out their different points of view on the study topic and thus answer the research question. All interviews were conducted in Italian. Before starting the interview, each participant was asked to complete a written informed consent form and answer some socio-demographic questions. The research was approved by the Bioethics Committee of the University of Turin. The study was conducted according to the ethical recommendations of the Declaration of Helsinki and the American Psychological Association (APA) standards for the treatment of volunteers. The data were collected in accordance with the Research Code of Ethics of the Italian Psychology Association.

### 2.2. Participants

Participants were 40 students enrolled at universities located in Northwestern Italy with an average age of 21.8 years (range: 19–25). In total, 50% identified as female at birth and 35% were enrolled in Social Sciences courses.

Another 15% were enrolled in Humanities courses, 20% in Medical and Pharmaceutical Sciences, 15% in Mathematical and Physical Sciences, 5% in Fine Arts, 5% in Conservatory Studies, and 5% in the Polytechnic School. A total of 67% of the participants were currently in a relationship, while 27% have had a partner in the past; 7% had never been in a relationship.

### 2.3. Data Analysis

The interviews were coded by two adequately trained research assistants who did not conduct the interviews. During the initial coding phase, the research assistants carefully read and re-read the transcripts to familiarize themselves with the data, and generated initial codes inductively by highlighting significant statements and recurring patterns. These codes were then discussed with the two researchers who conducted the interviews, and a consensus was reached on code definitions and their application across the dataset. The coding process was manual, without the use of qualitative analysis software. Following this, the researchers organized the codes into subthemes and broader themes, which were reviewed and refined through discussion in light of the existing literature, following a collaborative and reflective approach. To analyze the data, the researchers used the reflective thematic analysis approach (RTA) ([8]; [10]). In line with the principles of reflexive thematic analysis, themes were actively constructed through an iterative process of deep engagement with the data. After an initial phase of inductive coding, the research team systematically reviewed and discussed the codes to identify patterns of shared meaning. Codes with similar conceptual content were clustered around central organizing ideas, forming candidate themes. These themes were then refined, merged, or redefined through repeated comparison with data excerpts and collaborative reflection, ensuring they captured rich and multifaceted interpretations relevant to the research aims.

During interviews and analysis process, the two researchers repeatedly reflected on and discussed their own positionality as white Western women investigating a topic that statistically affects women more than men. Furthermore, in the initial coding phase two psychology students in training supported the researchers in identifying themes and subthemes. Their involvement made it possible to integrate, during the analysis process, the perspective of university young adult women, the target of the research, offering a viewpoint less influenced by the approaches of the researchers who conducted the interviews.

Each participant was assigned a code (P1, F/M = participant 1, female/male). Their contributions offered additional perspectives from young students in the same age group as participants, helping to balance potential researcher bias and integrate alternative viewpoints into the analytic process.

## 3. Results

From the data analysis, the following three themes emerged: (1) from visible to invisible violence; (2) the causality of gender-based violence—temperament or stereotypes?; and (3) using social media for relationships or for violence (see Table 1).

### 3.1. From Visible to Invisible Violence

This theme reflects how participants conceptualize the different forms of gender-based violence, moving from the more visible physical and sexual violence to less visible psychological, homophobic, socio-economic, and online violence.

#### 3.1.1. Physical Violence

Both male and female participants identified physical violence as the most visible and easily recognizable form. Examples frequently mentioned included hitting, punching, slapping, up to rape and femicide:
“*Physical violence is perhaps the most obvious one, meaning the easiest to recognize—like actual physical abuse, hitting, punching, etc.*”(P1, F)
“*All those actions that involve hitting the other person, or even just grabbing them forcefully by the wrists, pulling their hair… basically anything that causes physical harm.”*(P30, M)

#### 3.1.2. Psychological Violence

Although psychological violence was generally identified as secondary, many participants demonstrated a deep understanding of this form of abuse during the interviews, listing various types and emphasizing how it is
“*harder to recognize and, in my opinion, harder to stop.*”(P5, F)

Among the psychological forms of violence identified by participants there were manipulation, blackmail, control, deprivation of personal freedom, the imposition of one’s viewpoint on another, and verbal violence. The latter includes insults but also the intentional belittling of the other person.
“*But I perceive control in that way—when someone tries to impose their will on you.*”(P9, M)
“*Another form of violence is definitely negative comments that serve no constructive purpose and only make you more and more submissive to the other person.*”(P12, F)

A notable observation was made about psychological violence experienced by men, such as insults intended to undermine masculinity:
“*It often happens that one man tells another, ‘You’re a girl, you’re acting like a little girl’ to demean him.*”(P5, F)

Some participants also emphasized that psychological violence can be perpetrated regardless of gender, whenever an individual exploits their status to cause psychological harm:
“*Gender-based violence occurs when a man or a woman takes advantage of their role to inflict psychological harm.*”(P30, M)
“*In my opinion, gender-based violence also includes psychological aspects—words, phrases—and it affects all genders. It’s not something exclusive to just one gender.*”(P23, F)

#### 3.1.3. Homophobic Violence

Several participants, including male participants identifying as LGBTQ+, reported experiences or awareness of homophobic violence, illustrating how violence affects diverse sexual orientations:
“*It happened while I was with my boyfriend in a public place, somewhere relatively calm. Someone approached us and started yelling aggressively.*”(P9, M)

#### 3.1.4. Socio-Economic Violence

This less frequently mentioned form concerned economic dependency, financial control, and structural undervaluation of women’s work and competencies:
“*In many families, the husband is the one who provides for the household, and this makes the woman dependent on him in every aspect.*”(P25, M)
“*Many women don’t have money of their own; they often have to share finances with a man, which means they cannot achieve economic independence.*”(P19, F)

Structural inequality was highlighted, for instance, regarding gender pay gaps and professional recognition:
“*I know that, nowadays, in Italy, there is still a great disparity between men and women when it comes to salary.*”(P19, F)

Meanwhile, another participant refers to a more structural form of violence, which is perceptible in every domain where women’s competencies are undervalued:
“*In many households, it’s still expected that the woman does the cooking, but all the top chefs are men.*”(P21, F)

#### 3.1.5. Online Violence

Finally, the analysis of the interviews shows that only two participants mention online violence as one of the possible types of abuse. Despite participants showing a good understanding of physical, sexual, and psychological violence, when it comes to online violence, the two participants made a more general reference, listing examples such as stalking, cyberbullying, and negative comments under posts:
“*Then we have digital violence, like stalking, cyberbullying.*”(P33, F)
“*There are now quite a few videos circulating on these topics, and when the comments are opened, some of them are really disturbing on social media.*”(P39, F)

### 3.2. The Causality of Gender-Based Violence: Temperament or Stereotypes?

This theme explores participants’ perceptions of the roots of gender-based violence, distinguishing between individual predispositions and socio-cultural causes.

#### 3.2.1. Individual Predisposition

Some participants attributed violence to individual violent natures or psychological traits, including impulsivity and aggression:
“*At its core, there must always be a predisposition in the individual towards certain behaviors: it’s rare that someone who commits gender-based violence isn’t someone who already exhibits these behaviors in other contexts.*”(P30, M)
“*Aggressiveness, in the sense of acting impulsively, letting go of inhibitions, without taking a breath and acting impulsively, hitting a vulnerable part consciously, in other words, being aggressive in the context of gender in a deliberate way, essentially fueling that asymmetry.*”(P18, F)

Biological differences are also mentioned as a possible cause of gender-based violence.

Other participants pointed to biological differences between men and women, with one participant mentioning that women are often physically weaker than men, which can contribute to their vulnerability:
“*I think about this difference that often exists biologically between men and women, where unfortunately, a woman is usually less physically strong than a man.*”(P15, F)

Others recognize violence as a form of revenge, rooted in the pain perceived by the perpetrator:
“*There are people who, when faced with unpleasant situations, want to reflect that pain onto the one who caused it, so it becomes a sort of revenge.*”(P29, M)

Many participants also identify emotional factors such as jealousy, envy, fear of abandonment, and hatred as causes of violence. Some give examples of extreme jealousy leading to violence, including physical aggression and even murder:
“*There are more emotional causes, for example, many men commit physical violence and some even end up killing their partner because of jealousy.*”(P25, M)
“*He doesn’t want his partner to go out with friends or other people because he’s afraid of abandonment.*”(P33, F)

Others connect envy to the development of violence, with a progression from envy to hatred, and eventually to violent behaviors:
“*It starts from envy, and then that idea transforms into a kind of hatred toward another person. Perhaps from that step of hatred, true violence develops. Maybe in the beginning, it’s just envy, control, and a bit of jealousy linked to envy.*”(P3, M)

Some participants attribute violence to the desire for control, particularly among men, suggesting that this desire stems from psychological issues:
“*Probably the subtypes of violence stem from psychological problems in the male figure, who may have a need for control.*”(P4, F)

Other participants attribute violence to a lack of self-control, where stress and impatience can lead to aggression:
“*In certain situations, stress and patience can exceed a limit, and it becomes difficult to remain the same way: I consider myself a calm and composed person, but there are times when my patience exceeds its limit, and it’s hard to stay calm and collected as I always have been.*”(P29, M)

Finally, some participants link violence to unresolved trauma or difficult pasts, suggesting that violence can arise from internal fractures, particularly unresolved childhood traumas:
“*Those who commit violence, in general, always have an inner fracture… they may have had a difficult past, some dark motivations for what they do.*”(P28, M)
“*There may also be some unresolved trauma, perhaps sexual trauma. I think unresolved trauma in early age can resurface as hatred toward other genders.*”(P22, F)

#### 3.2.2. Socio-Cultural Causes

More extensively, participants identified cultural roots, gender norms, and power imbalances, as causes of violence:
“*When growing up, there’s always the belief that there is a dominant role, especially in culture, which only creates the male.*”(P10, F)
“*Male feel more powerful than the woman and therefore feels free to treat her in a certain way*”(P16, F)

Complementary to this line of thought, two participants argue that violence can stem from a sense of inferiority felt by men.
“*It could be that the male actually feels inferior to the female, and this leads him to various forms of violence, up to femicide.*”(P16, F)

Other causes, according to the participants, are related to role expectations and stereotypes that are so deeply rooted in the history of societies becoming invisible and unknowingly passing down from generation to generation.
“*For example, my boyfriend has a brother and a sister, and when it’s time to do the dishes, it’s always the sister who’s asked. But when there’s work that requires more strength—though not too much, so a girl can easily do it—the boy is called. I think this is something built by society.*”(P11, F)
“*It’s precisely because we’re talking about an imbalance in roles. Often, women are seen as a sort of weak form in the eyes of men, and so I see it as an imbalance within these contexts, both social and work-related. I saw this in my boyfriend’s family because his mother quit her job to take care of the child, more out of economic necessity, of course, but also because she has to take care of the house, cook, and clean.*”(P4, F)

The victim-blaming attitudes and justifications are reflected in the narrative of one participant, who shared how a personal experience that traumatized her, attributing the perpetrator’s behavior to certain sexual expectations with which he was raised:
“*He also expected certain things from me that I wasn’t ready to give him, but not in a bad way. I mean, he’s truly a good person, very calm, but he grew up with expectations about how the sexual life should be, which traumatized me somewhat in that regard.*”(P5, F)

Another participant, unwittingly influenced by an internalized stereotype, interprets the violence as an abusive attitude from men towards the vulnerability and sensitivity of women.
“*We women are much more sensitive to emotions, and so when faced with a violent man or a violent person, we try to focus on the positive aspects. This doesn’t mean we are weak, but this behavior can work against us, as a man may exploit our vulnerability.*”(P12, F)

One participant also addresses the role of bystanders in violence, asserting that one can contribute to violence simply by remaining silent.
“*One can contribute to violence simply by remaining silent, because silence means doing nothing, and I am convinced that neutrality means siding with the oppressor in many situations, if not all.*”(P24, F)

Lastly, a participant emphasizes that justifying and trivializing violence is also a form of violence.
“*Not only from those who commit it, but also from those who justify it. Failing to give it the attention it deserves.*”(P38, F)

### 3.3. Using of Social Media as a Tool for Building Relationships or Perpetrating Violence

This theme marks a critical transition from participants’ theoretical understanding of various forms of gender-based violence and their perceived causes to the more personal, lived experiences of violence. While both male and female participants demonstrated awareness of different types of violence in abstract or general terms, it is within this theme, centering on social media use and related experiences, that women predominantly report actual episodes of victimization. This section encompasses participants’ perceptions of social media both to facilitate social and romantic relationships and as a context in which violence, particularly online harassment and coercion, is experienced. In this regard, it is interesting to note that when participants were asked to list the types of violence they knew, online violence was the only type that was not deeply explored. However, when asked to speak about experiences related to online harassment, many had personal stories to share. This underscores a crucial shift from cognitive representations to emotional and experiential realities concerning gender-based violence in digital contexts.

Regarding the use of social media as a tool for meeting people, some participants highlight its strengths, describing social media as a tool to simplify encounters, express oneself, and as useful for more introverted individuals or even for protecting oneself from public rejection.
“*Reflecting on it, when using social media, any application, or in general, when you’re there talking, in the end, it’s different, but it’s simply a means to simplify an encounter.*”(P8, M)
“*I don’t know, maybe it could also be useful for people who are a bit shyer or don’t have the courage to meet people right away or are afraid of having a conversation of any kind in person.*”(P8, M)
“*Social media has allowed us to handle the disappointment of not being reciprocated without it being public. When a person approaches and isn’t accepted, yes, they face public rejection, and that can make someone a bit more intimidated*… *So I think it’s a positive thing.*”(P38, F)

Despite this, many girls report being subjected to forms of harassment, requests for photos in exchange for money, or having received intimate videos of other people without their consent.
“*There’s this man who keeps pestering me, even though I blocked him, he still keeps coming back, I’ve reported him several times, blocked him, and he keeps asking me, sending me posts.*”(P6, F)
“*Yes, yes, mainly on Instagram, from people I’ve never seen in my life, who maybe started following me and had the nerve to ask for intimate photos of mine in exchange for money.*”(P4, F)

The emotional experiences that emerge following these forms of violence can be dichotomous: initially, the prevalent reaction is the minimization of the situation, while later, with further processing, discomfort prevails.
“*So maybe at the time it just made me laugh because it was ridiculous, thinking about it afterward, and especially now that I’m older, I think, ‘What a disgusting thing,’ I mean, pull yourself together, you don’t send intimate photos to strangers, especially when you don’t even know the age of the strangers. Because in hindsight, I could have easily reported it, because I was a minor.*”(P13, F)

Some report feeling dirty: “I don’t know, dirty, even though I didn’t do anything wrong, but I have a feeling of discomfort.” (P10, F). Others, instead, feeling shocked: “So I had downloaded the Snapchat app through which you can reach people outside your circle of friends and phone contacts, and I received this photo from a stranger, of his penis, I was kind of shocked.” (P38, F)

Indifference or feeling proud, on the other hand, are emotional states reported mainly by young men.
“*It happened to me (that someone sent me photos), but well, in that case, personally, it didn’t cause me any problems. I just block the account and move on.*”(P8, M)
“*It happened to me that some girls sent me photos. From a male perspective, I’m proud.*”(P17, M)

Finally, within this theme, the issue of photo sharing is discussed. In this regard, many participants express concern about the possibility of losing control over a photo once it is shared.
“*On social media, it’s more difficult because I imagine the moment I share a photo, I lose control of it. So, if I somehow withdraw my consent, how do I trust the other person?*”(P18, F)

This fear is also present when the photo is shared with a partner.
“*When maybe I send a certain kind of photo thinking, ‘Well, it’s just for him.’ No one guarantees me that this photo will only stay with him and won’t be shown to the whole world.*”(P23, F)

Furthermore, there is a group of participants, mainly men, who believe that the person sending photos should take responsibility if their intimate material is shared afterward, and another participant refers to the sharing of photos without consent as incidents that happen very commonly:
“*If I send it without considering it, sure, it’s wrong for it to be shared, but I shouldn’t complain, so to speak, if it ends up being shared because the risk is there, period.*”(P28, M)
“*From a realistic point of view, these activities set the stage for a series of incidents that actually happen very frequently, too frequently to ignore.*”(P20, M)

## 4. Discussion

The study aimed to explore the social representations that university students have regarding gender-based violence, also based on their personal experiences, and to highlight the perception related to social media and how their (mis)use might be connected to gender-based violence.

### 4.1. Perception of Types of Violence

In line with the literature, the university students in this study appear to have a clear understanding of the types of violence, both in its more visible forms such as physical and sexual violence, and the more invisible ones, including emotional, psychological, and economic violence ([4]). Labeling certain acts or behaviors as ‘violence’ plays a crucial role in acknowledging their seriousness and, consequently, in determining their social (un)acceptability ([19]). However, despite young people being the group that uses social media the most, online violence was only briefly mentioned in a general way by a few participants, as highlighted by previous research ([18]; [38]).

### 4.2. Attributed Causes: Individual vs. Cultural

Regarding the possible causes of gender-based violence, some of the representations that emerged from the interviews point to a view of gender-based violence as an individual phenomenon, involving only a few people. Some participants associate gender-based violence with those who have a violent nature and act under uncontrollable impulses. Others attribute the causes to biological differences, where women are considered weaker than men. In this regard, the study by [19] ([19]) highlights how young people’s beliefs regarding biological gender differences, the construction of norms, and heterosexual gender relations significantly impact attitudes toward violence against women, contributing to its normalization and justification. Revenge and jealousy are additional potential causes identified by some participants, which, as previously noted in the literature, represent a form of justification for violence referenced by young adults ([6]; [19]; [49]; [54]). However, a literature review by [12] ([12]) also found that certain individual factors, such as depression and personality disorders, can serve as risk factors for intimate partner violence.

On the other hand, many of the participants identify the causes in a cultural framework where stereotypes, prejudices, and gender norms guide behavior. Several studies on ambivalent sexism and gender-based violence demonstrate how the latter is fueled by sexist attitudes, both hostile and benevolent, which normalize male control over women, underscoring the pivotal role of social beliefs and cultural norms ([13]; [33]; [48]; [46]). In addition, other participants explain the possible causes of violence by reporting personal experiences of victimization, through which it emerges that the participants themselves have internalized certain stereotypes and use mechanisms of justification that normalize and absolve the perpetrator and their behavior. These representations echo rape myths and violence justification, highlighting how people often attribute violence to uncontrollable impulses or external factors to absolve the aggressors and maintain the status quo. This reinforces the erroneous perception that violence is an individual issue rather than a systemic one ([1]).

### 4.3. Online Violence and Social Media

Finally, regarding social media, the data analyzed reveal that most participants have experienced forms of harassment, in line with quantitative studies on the subject ([18]). Women are the ones who report more episodes of victimization and, consequently, also experience more conflicting and negative emotions compared to men, but also normalization and minimization of the violence they have suffered, revealing deep-rooted gendered differences in emotional processing shaped by normative masculinity and femininity ideals ([19]). This contrasts with part of the literature indicating that men are more likely to endorse beliefs that justify violence against women, often underestimating the severity of certain violent behaviors directed at the female gender ([26]; [56]; [58]). In our study, women were more likely to recognize and report negative emotional experiences. This finding reflects well-documented gendered patterns in emotional expression: it is generally more socially acceptable for women to display emotions linked to vulnerability, whereas men are discouraged from expressing emotions perceived as weakness, as these are often stigmatized and associated with lower competence ([29]). Research adopting biopsychosocial and social learning perspectives shows that such differences emerge gradually through socialization processes, whereby children are encouraged to adopt gender-role consistent emotional behaviors ([14]). Gendered emotional standards, whether explicit or subtle, form the core of masculine and feminine identities. They shape how individuals acquire and maintain gender identity by guiding gender-coded emotional values and behaviors, influencing how emotions are experienced and interpreted ([25]).

Furthermore, the [21] ([21]) has highlighted how women tend not to recognize their lived experiences as crimes due to low awareness and sensitivity towards the phenomenon. On another point of view, these types of victimization take places also within intimate partner relationships, where the literature highlights the complex interplay between dependency, both economic or emotional, and the misinterpretation of violence ([7]).

Finally, many participants, both women and men, when asked about sharing photos without consent, referred to a sort of responsibility that the victim must bear. According to the participants, a person who chooses to share their intimate photos with someone else becomes guilty when such material is disseminated without their consent. This topic has generated extensive debate in the literature on victim-blaming, where the victim is often considered partially responsible for what happened, as if they could or should have prevented it. However, contrary to what emerged from this research, the literature highlights how men are more likely than women to endorse rape myths and to blame the victims of sexual violence ([16]). This attitude, however, only produces secondary victimization, with further psychological impact on the victim ([23]; [40]). Therefore, social media serves as both facilitators of social connection and arenas where traditional power disparities and gender-based control are reproduced and sometimes exacerbated. Consistent with [34]’s ([34]) emphasis, our findings highlight the critical importance of differentiating between online and offline forms of gender-based violence to fully appreciate the complexity and intersectionality inherent in these experiences. The data reveal that while offline violence remains more visible and commonly acknowledged, online violence presents unique dynamics and challenges that intersect with social media use, gender norms, and digital cultures.

## 5. Conclusions

Although this work allowed the viewpoint of college students to emerge, there are some limitations. First among them, the use of a qualitative instrument does not allow for the generalizability of the results. Another limitation is that the participants were university students, thus not representative of young adults in general. Finally, given participation was on a voluntary basis it is more likely that the participants who decided to participate had a greater basic sensitivity to this issue. Despite these limitations, this research brought out the views of university students who, using the interview, were free to use their own narrative expressions and narrate their representations and experiences related to the topic under investigation. Furthermore, the results reveal that young adults are very aware, at a theoretical level, of “offline” physical, psychological, and verbal gender-based violence and its effects, while they do not give much consideration to online violence, despite often being victims of it, as revealed by their accounts. They have had different experiences with it, accompanied by contrasting emotional responses. For example, regarding the nonconsensual dissemination of intimate images, many attribute some of the responsibility to the victim, demonstrating how victim-blaming is still a very widespread phenomenon, especially concerning online violence. Therefore, the results of this research highlight the need to develop prevention programs focused on increasing awareness and providing young people with more tools to identify when they have been victims of violence or bystanders, both online and offline, and to process the emotional experiences associated with such events. Furthermore, future research should quantitatively explore online and offline violence normalization and extend beyond academic contexts to capture broader young adult experiences.

## Figures and Tables

**Table 1 behavsci-15-01373-t001:** Themes that emerged from the semi-structured interviews.

*Theme*	*Sub-Theme*
*From visible to invisible violence*	physical
	sexual
	psychological
	homophobic
	socio-economic
	online
*The causality of gender-based violence: temperament or stereotypes?*	individual predisposition
	socio-cultural causes
*Using social media as a tool for building relationships or perpetrating violence*	tool for building relationships
	requests to share intimate photos: personal and emotional experiences

## Data Availability

The data that support the findings of this study are available from the corresponding author, [M.T.], upon reasonable request. The data are not publicly available due to privacy concerns and the potential risk of compromising participant confidentiality.

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
