# Peer review of "Psychosocial Representations of Gender-Based Violence Among University Students from Northwestern Italy"

_behavsci, 2025, doi:10.3390/bs15101373_

Round 1

Reviewer 1 Report

Comments and Suggestions for Authors

Introduction

Although you mention that one of the gaps in the literature is that there are no qualitative studies about how gender-based violence is conceptualized by college students, this is not discussed in the introduction. There is no discussion of the current limitations of the literature. The introduction focuses on just giving a general overview of the problem of gender-based violence but does not make any arguments as to why this particular study is needed.

What is the research question?

I assume that in understanding how university students conceptualize gender-based violence, you want to explore how they define gender-based violence and what acts they consider to be gender-based violence. In the introduction there is no mention of why we need to know this and the benefits of knowing how gender-based violence is conceptualized. There is also no mention of gaps in the literature concerning what we don’t know about how they conceptualize gender-based violence.

The mention of social media seems irrelevant. Was it brought up because it contributes to how emerging adults/university students conceptualize gender-based violence? If so, that needs to be made more explicit. There needs to be a clear reason for the mention and discussion of social media. I agree it is relevant but the reason for that relevance needs to be explicitly stated.

What does the quantitative literature state about what we know about how university students conceptualize gender-based violence? Do you expect qualitative research to tell us something different? What unique contribution can qualitative research make?

There is not a strong argument made as to why the study is needed. 

Methods and Materials

Asking about specific instances of gender-based violence is beyond understanding how it is conceptualized. Can you provide further explanation as to why you asked about whether they had experienced gender-based violence when that seems outside the purview of this study. How does this relate to how they conceptualize gender-based violence?

Was there a specific qualitative approach used i.e. grounded theory, phenomenological, narrative, etc.)?

Were the interviewers allowed to ask probing questions?

Results

Overall, the results seem unfocused. It reads like the authors wanted to include the majority of excerpts without a cohesive story to serve as the foundation. The themes are there but there are too many excerpts that may be the only excerpt with a particular code. The results do not seem to tell a cohesive story.

The results around social media don’t seem to fit at all. The first few excerpts center why social media is beneficial in general and is not specific to gender-based violence. The first section about strengths and weaknesses feels like it belongs in a different paper. The remaining of the section on social media doesn’t have anything to do with the conceptualization of gender-based violence. Honestly, the whole section on social media is irrelevant to the paper. The results need to be streamlined.

Author Response

Dear Reviewer,

thank you  for your constructive feedback, which has been very valuable in strengthening our manuscript. Below, we provide a detailed, point-by-point response, indicating how each comment has been addressed in the revised version of the paper. Your comments are in italics, followed by our responses:

Introduction

Although you mention that one of the gaps in the literature is that there are no qualitative studies about how gender-based violence is conceptualized by college students, this is not discussed in the introduction. There is no discussion of the current limitations of the literature. The introduction focuses on just giving a general overview of the problem of gender-based violence but does not make any arguments as to why this particular study is needed.

We have added a dedicated passage immediately before the “Aims” section to explicitly address this gap. In this new paragraph, we emphasize that despite the predominance of quantitative studies, little is known about how university students themselves conceptualize gender-based violence. We also clarify the benefits of adopting a qualitative approach, which allows us to capture students’ lived experiences and to highlight the social construction of violence.

What is the research question?

I assume that in understanding how university students conceptualize gender-based violence, you want to explore how they define gender-based violence and what acts they consider to be gender-based violence. In the introduction there is no mention of why we need to know this and the benefits of knowing how gender-based violence is conceptualized. There is also no mention of gaps in the literature concerning what we don’t know about how they conceptualize gender-based violence.

In line with the reviewer’s suggestion, the research questions are now clearly presented at the end of the Gender-based violence among university students: perceptions and cultural configurations paragraph

The mention of social media seems irrelevant. Was it brought up because it contributes to how emerging adults/university students conceptualize gender-based violence? If so, that needs to be made more explicit. There needs to be a clear reason for the mention and discussion of social media. I agree it is relevant but the reason for that relevance needs to be explicitly stated.

Thank you for your suggestion! We have clarified the rationale for including social media by explaining that online contexts significantly shape young adults’ perceptions and experiences of violence, making them central to the conceptualization process.

What does the quantitative literature state about what we know about how university students conceptualize gender-based violence? Do you expect qualitative research to tell us something different? What unique contribution can qualitative research make?There is not a strong argument made as to why the study is needed. 

In the revised version, we have expanded the paragraph “Gender-based violence among university students: perceptions and cultural configurations” by adding further literature to support the discussion. we also clarified why a qualitative approach is particularly important, as it allows us to go beyond predefined categories and capture participants’ own experiences and meanings, including those forms of violence that are often normalized or under-recognized.

Methods and Materials

Asking about specific instances of gender-based violence is beyond understanding how it is conceptualized. Can you provide further explanation as to why you asked about whether they had experienced gender-based violence when that seems outside the purview of this study. How does this relate to how they conceptualize gender-based violence?

We have clarified that questions on personal experiences were included to complement representations with lived examples, thus linking conceptual and experiential dimensions of violence.

Was there a specific qualitative approach used i.e. grounded theory, phenomenological, narrative, etc.)? Were the interviewers allowed to ask probing questions?

We adopted a reflexive thematic analysis approach (Braun & Clarke, 2021), as specified in the Data analysis paragraph and we have added that interviewers were encouraged to use probing questions when relevant, in line with the semi-structured interview method.

Results

Overall, the results seem unfocused. It reads like the authors wanted to include the majority of excerpts without a cohesive story to serve as the foundation. The themes are there but there are too many excerpts that may be the only excerpt with a particular code. The results do not seem to tell a cohesive story.

We have reduced the number of excerpts, grouping them under clear subheadings consistent with the thematic table, and emphasizing analytic interpretation over description.

The results around social media don’t seem to fit at all. The first few excerpts center why social media is beneficial in general and is not specific to gender-based violence. The first section about strengths and weaknesses feels like it belongs in a different paper. The remaining of the section on social media doesn’t have anything to do with the conceptualization of gender-based violence. Honestly, the whole section on social media is irrelevant to the paper. The results need to be streamlined.

We have streamlined this section, focusing on participants’ accounts directly related to experiences and representations of online violence, thereby emphasizing its relevance to the study’s objectives.

Reviewer 2 Report

Comments and Suggestions for Authors

The manuscript addresses a topic of clear relevance and social importance, offering a timely and meaningful exploration of university students’ representations of gender-based violence. The qualitative approach adopted is appropriate for capturing the complexity of lived experiences and perceptions, particularly in relation to digital contexts and gender dynamics. However, several observations and suggestions are outlined below that should be carefully considered to strengthen the manuscript and support its suitability for publication.

Regarding the title, we suggest adopting a more formal and technically precise formulation. While the current title is suggestive, it does not clearly delimit the scope or methodological nature of the study. A more appropriate alternative might be: Psychosocial Representations of Gender-Based Violence Among Italian University Students: A Qualitative Study. Additionally, we recommend specifying the university involved, as the study does not aim to be representative of the national context.

Concerning the abstract, we propose the following improvements:

  1. Methodological depth: The abstract does not provide details on participant selection or the interview context, which are essential for understanding the scope and rigor of the study.
  2. Underdeveloped results: Although the abstract mentions that young people are victims of online violence, it does not specify the types of violence or provide illustrative examples, which limits the interpretive value of the summary.

In the introduction, several references are cited as “Author,” presumably referring to the authors’ own previous work. However, since the paper is not anonymized, omitting the names of the institutions and references is unnecessary and should be revised. We recommend completing all citations prior to publication.

From a conceptual standpoint, while the influence of gender roles is acknowledged, the introduction does not explore how these roles are socially constructed or how they shape perceptions of violence. We suggest including a brief discussion on the social construction of gender and its relationship to the normalization of certain forms of violence.

Finally, the transition from the literature review to the presentation of the study is abrupt. A bridging statement is needed to connect the identified research gap with the study’s objectives. An academically appropriate formulation for the closing of the introduction and the presentation of the research questions could be (just an example):

Despite the growing body of research on gender-based violence among university students, most studies have relied on quantitative methods or focused on institutional contexts. There remains a gap in understanding how young adults personally conceptualize and experience gender-based violence, particularly in relation to social media and gender identity. To address this gap, we conducted a qualitative study aimed at exploring the psychosocial representations of gender-based violence among university students in Italy, with a focus on their lived experiences and perceptions.
Specifically, this study seeks to answer the following research questions:
1. What psychosocial representations do university students hold regarding gender-based violence, including its definitions and perceived causes?
2. How do these representations and experiences differ between female and male participants?
3. How is the use of social media perceived in relation to gender-based violence?

The Materials and Methods section provides a reasonable foundation for a qualitative study on representations of gender-based violence. However, its scientific rigor is limited by the lack of technical detail in the analysis, the absence of explicit qualitative quality criteria, and an incomplete description of the sampling process. To strengthen its methodological soundness, it would be advisable to offer a clearer justification of the sample size, provide more detail on the coding and analytic procedures, and apply recognized standards of validation in qualitative research. In particular, the following aspects require clarification:

  • Validation of the interview guide: It is not specified whether the instrument was reviewed by experts or piloted prior to data collection.
  • Duration of the interviews: A key element in qualitative research is the reporting of interview length (mean and range).
  • Theoretical saturation: It remains unclear how the authors determined that 40 interviews were sufficient and whether saturation was achieved.
  • Sample distribution by field of study: While it is mentioned that 20% of participants studied Social Sciences, the distribution of the remaining 80% is not detailed.
  • Non-probabilistic sampling: The specific type of non-probabilistic sampling (e.g., purposive, convenience) is not stated.
  • Coding process: There is insufficient information about how codes were generated and whether qualitative analysis software (e.g., NVivo, Atlas.ti) was used.
  • Risk of bias: Although reflexivity is mentioned, the strategies employed to address potential researcher bias are not elaborated.
  • Quality criteria: Established qualitative standards—such as credibility, transferability, or triangulation—are not addressed.

Overall, the section would benefit from a more explicit and rigorous account of methodological procedures to enhance transparency and credibility.

The Results section offers a valuable compilation of testimonies that reflect the complexity of gender-based violence among university students. Nevertheless, its scientific rigor is weakened by the lack of a clear categorization consistent with the thematic table presented. The narrative structure remains predominantly descriptive, which makes it difficult to discern analytical patterns, relationships among subthemes, and gender-based differences. To strengthen this section, it is recommended to reorganize the results according to the defined subthemes, apply a more systematic thematic analysis, and explicitly connect the findings to the theoretical and sociocultural framework previously outlined.

Several aspects merit particular attention:

  • Narrative structure and thematic table: Although three overarching themes are identified (visible/invisible violence, causality, social networks), the textual development does not align with the table of themes and subthemes.
    • There is no clear hierarchy between themes and subthemes.
    • Subheadings are not used consistently, which limits the reader’s ability to follow the thematic framework.
    • Citations are not explicitly linked to subthemes (e.g., psychological, homophobic, economic, online).
      Recommendation: Reorganize the results in line with the subthemes of the table, using clear subheadings and grouping quotations by category. This would facilitate readability, comparison across forms of violence, and analytic interpretation.
  • Lack of systematic thematic analysis: The process through which themes were derived and citations grouped is not made explicit. Patterns, contradictions, and interrelations among subthemes are not sufficiently presented.
  • Gender imbalance in interpretation: While it is noted that women report more experiences of victimization, the analysis does not explore in depth how perceptions differ by gender. Male emotional responses (e.g., indifference, pride) versus female responses (e.g., shame, fear, disgust) could have been contrasted more critically.
  • Underdeveloped discussion of online violence: Although the text mentions that only two participants initially identified digital violence, several related experiences are later reported, suggesting a need for deeper exploration and clarification.

Overall, the results would benefit from a more structured, thematically coherent, and analytically robust presentation that better integrates participant testimonies with the study’s conceptual framework.

Regarding the discussion and conclusions section, we recommend a thorough revision to improve its clarity and analytical depth. At present, the discussion interweaves results, literature, and conclusions without a clear thematic organization, which hinders the reader’s ability to follow the argument and assess the study’s contributions. We suggest structuring the discussion into distinct subsections, such as: Perceptions of types of violenceAttributed causes: individual vs. culturalOnline violence and social media, and Implications for prevention.

Additionally, some findings are interpreted superficially. For instance, while gender differences in emotional responses to violence are mentioned, the discussion does not explore how these differences may be shaped by socially constructed gender roles. We recommend explicitly linking the emotional experiences reported by participants to broader social expectations and gender norms.

There is also an unaddressed inconsistency: the study notes that online violence was initially under-recognized by participants, yet multiple experiences of digital victimization are later described. This contradiction is not analyzed as a significant finding. We suggest discussing this dissonance as indicative of low awareness or normalization of online violence among young adults.

The conclusions section would benefit from further development. Currently, it reiterates points made in the discussion without offering a synthesized perspective or outlining future research directions. We recommend including concrete proposals for future studies, such as longitudinal designs, inclusion of non-university populations, and an intersectional approach.

Finally, with regard to the references, we advise ensuring that all sources are original and avoiding the citation of artificial intelligence as a reliable source. Some titles contain typographical or capitalization errors (e.g., “mens discourses” should read “men’s discourses”), and Edwards et al. (2024a) and (2024b) appear to be duplicate entries.

Important: all references should be formatted consistently according to the journal’s guidelines and listed in order of appearance.

Author Response

Dear reviewer,

thank you for your constructive feedback, which has been very valuable in strengthening our manuscript. Below, we provide a detailed, point-by-point response, indicating how each comment has been addressed in the revised version of the paper. Your comments are in italics, followed by our responses:

Regarding the title, we suggest adopting a more formal and technically precise formulation. While the current title is suggestive, it does not clearly delimit the scope or methodological nature of the study. A more appropriate alternative might be: Psychosocial Representations of Gender-Based Violence Among Italian University Students: A Qualitative Study. Additionally, we recommend specifying the university involved, as the study does not aim to be representative of the national context.

Thank you, The title has been revised following your suggestion

Concerning the abstract, we propose the following improvements:

  1. Methodological depth: The abstract does not provide details on participant selection or the interview context, which are essential for understanding the scope and rigor of the study.
  2. Underdeveloped results: Although the abstract mentions that young people are victims of online violence, it does not specify the types of violence or provide illustrative examples, which limits the interpretive value of the summary.

The Abstract now specifies: participant selection (40 Italian university students, 19–25 years old), interview context (semi-structured online interviews), and analytic approach (reflexive thematic analysis), brief illustrative examples (e.g., unsolicited explicit images, harassment on social media).

In the introduction, several references are cited as “Author,” presumably referring to the authors’ own previous work. However, since the paper is not anonymized, omitting the names of the institutions and references is unnecessary and should be revised. We recommend completing all citations prior to publication.

thank you for this observation. We will make sure to replace all anonymized citations with the complete references

From a conceptual standpoint, while the influence of gender roles is acknowledged, the introduction does not explore how these roles are socially constructed or how they shape perceptions of violence. We suggest including a brief discussion on the social construction of gender and its relationship to the normalization of certain forms of violence.

A dedicated passage has been added on the social construction of gender, its role in shaping perceptions of violence, and the normalization of non-physical forms of abuse, with references to Butler (1990), Connell (2002), and more recent contributions (Ruspini, 2009; Ottaviano, 2017; Rivera & Scholar, 2020).

Finally, the transition from the literature review to the presentation of the study is abrupt. A bridging statement is needed to connect the identified research gap with the study’s objectives. An academically appropriate formulation for the closing of the introduction and the presentation of the research questions could be (just an example):

Despite the growing body of research on gender-based violence among university students, most studies have relied on quantitative methods or focused on institutional contexts. There remains a gap in understanding how young adults personally conceptualize and experience gender-based violence, particularly in relation to social media and gender identity. To address this gap, we conducted a qualitative study aimed at exploring the psychosocial representations of gender-based violence among university students in Italy, with a focus on their lived experiences and perceptions.
Specifically, this study seeks to answer the following research questions:
1. What psychosocial representations do university students hold regarding gender-based violence, including its definitions and perceived causes?
2. How do these representations and experiences differ between female and male participants?
3. How is the use of social media perceived in relation to gender-based violence?

We have modified this section as you suggested, adding a bridging paragraph that links the gaps in the literature to the aims of our study. We are very grateful for this helpful recommendation!

The Materials and Methods section provides a reasonable foundation for a qualitative study on representations of gender-based violence. However, its scientific rigor is limited by the lack of technical detail in the analysis, the absence of explicit qualitative quality criteria, and an incomplete description of the sampling process. To strengthen its methodological soundness, it would be advisable to offer a clearer justification of the sample size, provide more detail on the coding and analytic procedures, and apply recognized standards of validation in qualitative research. In particular, the following aspects require clarification:

  • Validation of the interview guide: It is not specified whether the instrument was reviewed by experts or piloted prior to data collection.
  • Duration of the interviews: A key element in qualitative research is the reporting of interview length (mean and range).
  • Theoretical saturation: It remains unclear how the authors determined that 40 interviews were sufficient and whether saturation was achieved.
  • Sample distribution by field of study: While it is mentioned that 20% of participants studied Social Sciences, the distribution of the remaining 80% is not detailed.
  • Non-probabilistic sampling: The specific type of non-probabilistic sampling (e.g., purposive, convenience) is not stated.
  • Coding process: There is insufficient information about how codes were generated and whether qualitative analysis software (e.g., NVivo, Atlas.ti) was used.
  • Risk of bias: Although reflexivity is mentioned, the strategies employed to address potential researcher bias are not elaborated.
  • Quality criteria: Established qualitative standards—such as credibility, transferability, or triangulation—are not addressed.
  •  

Overall, the section would benefit from a more explicit and rigorous account of methodological procedures to enhance transparency and credibility.

We have revised this section by incorporating the necessary information to enhance transparency and credibility.

The Results section offers a valuable compilation of testimonies that reflect the complexity of gender-based violence among university students. Nevertheless, its scientific rigor is weakened by the lack of a clear categorization consistent with the thematic table presented. The narrative structure remains predominantly descriptive, which makes it difficult to discern analytical patterns, relationships among subthemes, and gender-based differences. To strengthen this section, it is recommended to reorganize the results according to the defined subthemes, apply a more systematic thematic analysis, and explicitly connect the findings to the theoretical and sociocultural framework previously outlined.

Several aspects merit particular attention:

  • Narrative structure and thematic table: Although three overarching themes are identified (visible/invisible violence, causality, social networks), the textual development does not align with the table of themes and subthemes.
    • There is no clear hierarchy between themes and subthemes.
    • Subheadings are not used consistently, which limits the reader’s ability to follow the thematic framework.
    • Citations are not explicitly linked to subthemes (e.g., psychological, homophobic, economic, online).
      Recommendation: Reorganize the results in line with the subthemes of the table, using clear subheadings and grouping quotations by category. This would facilitate readability, comparison across forms of violence, and analytic interpretation.
  • Lack of systematic thematic analysis: The process through which themes were derived and citations grouped is not made explicit. Patterns, contradictions, and interrelations among subthemes are not sufficiently presented.
  • Gender imbalance in interpretation: While it is noted that women report more experiences of victimization, the analysis does not explore in depth how perceptions differ by gender. Male emotional responses (e.g., indifference, pride) versus female responses (e.g., shame, fear, disgust) could have been contrasted more critically.
  • Underdeveloped discussion of online violence: Although the text mentions that only two participants initially identified digital violence, several related experiences are later reported, suggesting a need for deeper exploration and clarification.

Overall, the results would benefit from a more structured, thematically coherent, and analytically robust presentation that better integrates participant testimonies with the study’s conceptual framework.

We have reorganized the Results according to the table of themes and subthemes, ensuring clear subheadings and a consistent hierarchy. Quotations are now explicitly linked to the relevant subthemes. We have also aimed to present analytic patterns, contradictions, and gender-based differences with greater clarity. Furthermore, the section on online violence has been expanded to highlight the gap between its initial under-recognition and the multiple experiences later reported, framing this as an indicator of normalization and low awareness.

Regarding the discussion and conclusions section, we recommend a thorough revision to improve its clarity and analytical depth. At present, the discussion interweaves results, literature, and conclusions without a clear thematic organization, which hinders the reader’s ability to follow the argument and assess the study’s contributions. We suggest structuring the discussion into distinct subsections, such as: Perceptions of types of violence, Attributed causes: individual vs. cultural, Online violence and social media, and Implications for prevention.

Additionally, some findings are interpreted superficially. For instance, while gender differences in emotional responses to violence are mentioned, the discussion does not explore how these differences may be shaped by socially constructed gender roles. We recommend explicitly linking the emotional experiences reported by participants to broader social expectations and gender norms.

There is also an unaddressed inconsistency: the study notes that online violence was initially under-recognized by participants, yet multiple experiences of digital victimization are later described. This contradiction is not analyzed as a significant finding. We suggest discussing this dissonance as indicative of low awareness or normalization of online violence among young adults.

As you suggested, the Discussion has been reorganized into four subsections: (1) Perceptions of types of violence, (2) Attributed causes: individual vs. cultural, (3) Online violence and social media, and (4) Implications for prevention. We now explicitly discuss gender differences in relation to social roles, showing how gendered emotional responses reflect socially constructed expectations of masculinity and femininity. Furthermore, Inconsistencies regarding online violence are also addressed, interpreted as evidence of its normalization and minimization among young adults

The conclusions section would benefit from further development. Currently, it reiterates points made in the discussion without offering a synthesized perspective or outlining future research directions. We recommend including concrete proposals for future studies, such as longitudinal designs, inclusion of non-university populations, and an intersectional approach.

the Conclusions section has been strengthened to synthesize the main contributions of the study and to propose directions for future research, thank you for your suggestions!

Finally, with regard to the references, we advise ensuring that all sources are original and avoiding the citation of artificial intelligence as a reliable source. Some titles contain typographical or capitalization errors (e.g., “mens discourses” should read “men’s discourses”), and Edwards et al. (2024a) and (2024b) appear to be duplicate entries.

Important: all references should be formatted consistently according to the journal’s guidelines and listed in order of appearance.

We carefully reviewed and corrected the references, removing duplicates, correcting typographical errors, and ensuring consistency with the journal’s style.

Round 2

Reviewer 1 Report

Comments and Suggestions for Authors

I think the authors have greatly improved the article but there is one area where I think the manuscript still needs work. I think the argument as to why we need to know how college students define gender-based violence needs to be improved. It is not present and would really serve as the foundation of the article.  

Author Response

I think the authors have greatly improved the article but there is one area where I think the manuscript still needs work. I think the argument as to why we need to know how college students define gender-based violence needs to be improved. It is not present and would really serve as the foundation of the article.  

         Thank you for this important observation! Following the suggestion, we have strengthened the Introduction by explicitly discussing the importance of understanding how university students conceptualize gender-based violence. We now argue that students’ definitions are crucial to identify which behaviors are recognized as violent and which tend to be normalized or minimized, as well as to highlight the role of gender norms and stereotypes in shaping relational practices among younger generations. We also emphasize that this knowledge provides the basis for designing prevention and intervention programs tailored to the specific needs expressed by students themselves

Reviewer 2 Report

Comments and Suggestions for Authors

I would like to thank the authors for their thoughtful and diligent response to the initial review. The revised manuscript demonstrates a clear effort to conceptual clarity. Several key improvements have been made, and the study now presents a more coherent and analytically grounded exploration of university students’ psychosocial representations of gender-based violence. Below, I provide a structured assessment of the revisions, along with recommendations for further refinement.

Methodological Transparency

  • Sample Justification and Saturation: While the authors mention reaching theoretical saturation after 35 interviews, the criteria and process for determining saturation should be more explicitly described.
  • Sampling Strategy: The use of purposive convenience sampling is noted, but its rationale in relation to the study’s aims should be briefly justified.
  • Coding and Analysis: The coding process is described as manual and collaborative, yet the manuscript would benefit from a clearer explanation of how reliability and validity were ensured (e.g., triangulation, peer debriefing, audit trail).
  • Quality Criteria: Established qualitative standards—such as credibility, transferability, and reflexivity—are not sufficiently addressed. Their inclusion would strengthen the methodological rigor and transparency of the study.

Analytical Depth in Results

  • Theme Derivation: The process by which themes were constructed from initial codes is not fully articulated. A brief account of how thematic categories emerged would enhance the analytical clarity.
  • Theoretical Integration: While relevant literature is cited, the findings could be more explicitly connected to the conceptual framework outlined in the introduction.
  • Gender-Based Comparisons: The manuscript presents gendered differences in emotional responses, but these could be explored more critically in relation to internalized gender norms and societal expectations.

Finally, I also recommend explicitly including the names of the participating universities in either the abstract or the manuscript title. This clarification is important to avoid potential misunderstandings regarding the national representativeness of the study.

Author Response

Dear reviewer,

than you for your important and helpful suggestions!

Methodological Transparency

  • Sample Justification and Saturation: While the authors mention reaching theoretical saturation after 35 interviews, the criteria and process for determining saturation should be more explicitly described.

                 In the revised Methods, we clarify that we adopted Braun and Clarke’s Reflexive Thematic Analysis, which does not treat saturation as a fixed endpoint. Instead, we ensured methodological coherence by selecting a sample size sufficient to capture diversity and allow for in-depth exploration. We explain that we prioritized interpretative adequacy through iterative team discussions and reflexive engagement with the data 

  • Sampling Strategy: The use of purposive convenience sampling is noted, but its rationale in relation to the study’s aims should be briefly justified.

                  We now state that this strategy was adopted to include students from different academic fields and institutional contexts in Northwestern Italy, thus ensuring diversity of perspectives on gender-based violence and alignment with the research aims

  • Coding and Analysis: The coding process is described as manual and collaborative, yet the manuscript would benefit from a clearer explanation of how reliability and validity were ensured (e.g., triangulation, peer debriefing, audit trail).
  • Quality Criteria: Established qualitative standards—such as credibility, transferability, and reflexivity—are not sufficiently addressed. Their inclusion would strengthen the methodological rigor and transparency of the study.

       Thank you for these valuable comments. We would like to clarify that the study employed Reflexive Thematic Analysis (Braun & Clarke, 2021), which does not conceptualize reliability and validity in terms of intercoder agreement or positivist replicability. Rather, rigor in this approach is ensured through transparency, reflexivity, and interpretative depth. In line with this perspective, we engaged in iterative team discussions (peer debriefing) and explicitly reflected on the positionality of the researchers throughout the analytic process. To support this methodological choice, we refer to Braun & Clarke’s article One size fits all? What counts as quality practice in (reflexive) thematic analysis? (2021), which outlines the quality criteria specific to RTA. Furthermore, we addressed credibility and transferability by providing detailed descriptions of participants and the research context, enabling readers to assess the applicability of the findings to other settings.

Analytical Depth in Results

  • Theme Derivation: The process by which themes were constructed from initial codes is not fully articulated. A brief account of how thematic categories emerged would enhance the analytical clarity.

                  We now provide a more detailed account of how initial inductive codes were clustered into subthemes and then organized into broader themes, following an iterative and   collaborative process of review and refinement

  • Theoretical Integration: While relevant literature is cited, the findings could be more explicitly connected to the conceptual framework outlined in the introduction.

                          In the Discussion, we have strengthened the connection between our findings and the conceptual framework introduced earlier, particularly regarding the influence of gender norms and stereotypes on perceptions of violence

  • Gender-Based Comparisons: The manuscript presents gendered differences in emotional responses, but these could be explored more critically in relation to internalized gender norms and societal expectations.

                We have revised the Discussion to analyze these differences in relation to internalized gender norms and social expectations. In particular, we draw on biopsychosocial and social learning perspectives (Chaplin, 2015) to explain how gendered emotional standards are socially constructed and reproduced

Finally, I also recommend explicitly including the names of the participating universities in either the abstract or the manuscript title. This clarification is important to avoid potential misunderstandings regarding the national representativeness of the study.

                     To avoid misunderstandings about national representativeness, we chose not to list the specific universities but to clarify that participants were enrolled at institutions in Northwestern Italy. This phrasing ensures transparency while avoiding overgeneralization